# Micro-Addition of Silver to Copper: One Small Step in Composition, a Change for a Giant Leap in Biocidal Activity

**DOI:** 10.3390/ma17040917

**Published:** 2024-02-16

**Authors:** Vitor G. Vital, Márcio R. Silva, Vinicius T. Santos, Flávia G. Lobo, Patrícia Xander, Rogéria C. Zauli, Carolina B. Moraes, Lucio H. Freitas-Junior, Cecíla G. Barbosa, Diogo S. Pellosi, Ricardo A. G. Silva, André Paganotti, Suzan P. Vasconcellos

**Affiliations:** 1Instituto de Ciências Ambientais, Químicas e Farmacêuticas—ICAQF, Universidade Federal de São Paulo—UNIFESP, Diadema 09913-030, Brazilpatricia.xander@unifesp.br (P.X.); rczauli@unifesp.br (R.C.Z.);; 2Department of Research and Development, Termomecanica São Paulo S.A., São Bernardo do Campo 09612-000, Brazil; 3Faculdade de Ciências Farmacêuticas, Universidade de São Paulo, São Paulo 05508-000, Brazil; 4Departamento de Microbiologia, Instituto de Ciências Biomédicas, Universidade de São Paulo, São Paulo 05508-900, Brazil; 5Instituto de Química, Universidade Federal do Paraná, Curitiba 81531-980, Brazil

**Keywords:** copper, silver, biocidal, H1N1 virus, alloys

## Abstract

The use of copper as an antimicrobial agent has a long history and has gained renewed interest in the context of the COVID-19 pandemic. In this study, the authors investigated the antimicrobial properties of an alloy composed of copper with a small percentage of silver (Cu-0.03% wt.Ag). The alloy was tested against various pathogens, including *Escherichia coli*, *Staphylococcus aureus*, *Candida albicans*, *Pseudomonas aeruginosa*, and the H1N1 virus, using contact exposure tests. Results showed that the alloy was capable of inactivating these pathogens in two hours or less, indicating its strong antimicrobial activity. Electrochemical measurements were also performed, revealing that the small addition of silver to copper promoted a higher resistance to corrosion and shifted the formation of copper ions to higher potentials. This shift led to a slow but continuous release of Cu^2+^ ions, which have high biocidal activity. These findings show that the addition of small amounts of silver to copper can enhance its biocidal properties and improve its effectiveness as an antimicrobial material.

## 1. Introduction

Copper has been recognized for its antimicrobial properties since ancient times [1]. This versatile element is a key constituent of many metalloenzymes and proteins that play important roles in electron transport and other metabolic processes [2]. As such, copper is considered an essential micronutrient for most, if not all, living organisms. Regarding microorganisms, their demand for copper is usually around 1–10 μmol/L. On the other hand, excess of copper ions is highly toxic to microorganisms.

In addition to its essential biological functions, copper exhibits potent antimicrobial activity. This property has been exploited for centuries, using copper to fabricate vessels and surfaces for water and food storage, and more recently, for the production of various medical instruments [3]. Recent studies have confirmed the efficacy of copper as an antimicrobial agent, with research indicating that copper surfaces can reduce bacterial contamination by up to 99.900% [1]. This is due to the ability of copper ions to disrupt the cell membranes of microorganisms, ultimately leading to their death [4]. Moreover, copper’s antimicrobial properties have been found to be particularly effective against antibiotic-resistant bacteria, making it an attractive alternative to traditional antibiotics in the fight against infections [5]. Copper is a remarkable element with intrinsic antimicrobial properties and essential biological functions. Its potential as an antimicrobial agent makes it a valuable tool in the fight against infectious diseases, especially those caused by antibiotic-resistant bacteria [1,3]. Although some microorganisms can develop come sort of copper resistance, prolonged exposure to copper is toxic to microorganisms [6].

With the emergence of new drug-resistant pathogens and increasing antibiotic resistance, metals and metal nanoparticles are being investigated as a potential solution to the problem of superbugs [7]. Among metal-based antimicrobial agents, silver-based compounds have been extensively studied. Silver ions have broad-spectrum antimicrobial activity against a wide range of bacteria, fungi, and viruses [8]. Additionally, silver nanoparticles (AgNPs) have been shown to exhibit enhanced antimicrobial activity compared to their bulk counterparts.

The mechanism of action of silver-based compounds against microorganisms is not fully understood, but it is thought to involve the release of silver ions from the metal (bulk or nanoparticle) to its proximities [9]. These ions can interact with the bacterial cell membrane, damaging it and disrupting cellular processes [10]. Additionally, silver ions can interact with cellular components such as DNA and proteins, leading to further damage and ultimately cell death.

The addition of silver to copper to create an alloy has been shown to enhance the biocidal activity of copper, making it a more effective antimicrobial material. This synergistic biocidal effect [11] has been demonstrated in several studies, in which copper-silver alloys can effectively reduce the viability of a variety of bacterial pathogens, including multidrug-resistant strains [1,3]. Additionally, the use of copper–silver alloys as antimicrobial materials has several advantages over traditional materials coated with antibiotics, including a reduced risk of inducing bacterial resistance and a lower potential for toxicity [12]. The antimicrobial properties of copper and silver have also been shown to reduce the rate of healthcare-acquired infections in hospital environments [3].

It is important to note that while the addition of silver to copper has been shown to enhance its biocidal properties, there is no current research on the effects of micro-additions of silver (<0.1%) to copper. Most studies on copper–silver alloys have focused on compositions containing large amounts of silver, typically 5–10% [1,11]. Therefore, the potential biocidal benefits of very low levels of silver in copper are not yet fully understood and warrant further investigation. Nevertheless, the effectiveness of copper-silver alloys against a wide range of pathogens has been well documented [13], and they remain a promising material for use in various applications, including in healthcare settings and water purification systems [3].

## 2. Materials and Methods

The copper alloys used in this study were kindly donated by Termomecanica S.A. (São Bernardo do Campo, Brazil), presenting 0.03 wt.% of silver. This alloy is defined by the Unified Numbering System for copper and copper alloys as C-10400 and classified by the U.S. Environmental Protection Agency as antimicrobial [14,15]. It was manufactured by casting in industrial induction furnaces, having been hot- and cold-rolled and heat-treated to the mentioned specifications. To evaluate their biocidal activity of the alloy, the alloys were tested under pathogens loads in established protocols. Metallic salts were used a control group. More details are available in the following topics.

The biocidal activity of the alloy against pathogens was determined by adapting an international protocol [16] and using the Japanese test method [17] as a foundation. The latter mentioned method is based on the recovery of viable cells after the exposure of the metallic surface to the microorganism. The copper alloy was exposed to four model pathogens, those being *Escherichia coli* (ATCC 8739), *Staphylococcus aureus* (ATCC 6538), *Pseudomonas aeruginosa* (ATCC 9027), and *Candida albicans* (CA INCQS 40006). The alloys samples were sterilized using ultraviolet light for thirty minutes after being submerged in ethanol (70% purity). After the sterilization of both sides, the disc-shaped samples were placed in Petri dishes then inoculated with 200 µL of a solution containing 10^8^ CFU/mL, with CFU an anacronym for colony-forming unit. The samples were then covered with a thin film of sterilized polyethylene. Immediately after the contract and at 0.5, 2, 4 and 24 h, samples were collected following the protocol to be described; during this time, the dishes were maintained at 35 °C and a relative humidity of 90% or higher to avoid death by dehydration of the cells. To collect the sample, the plastic film was removed from the alloy and the sample was washed with 10 mL of 0.1% peptone water. This washing water was then diluted to the desired concentrations to be cultivated in a solid, agar-based medium; finally, the viable cells were counted. To evaluate if the exposure to the surface affected the growth of the bacteria, a one-Sample *t*-test between the tested value and positive control was conducted using OriginPro^®^ 2021, and the value was considered to be statistically different with a *p*-value criterion of *p* ≤ 0.01.

The *C. albicans* cells’ viability and death were determined using flow cytometry before and after contact with the alloys. After the different treatments, the fungus was labeled with propidium iodide (Invitrogen, Waltham, MA, USA). Then, the yeasts were analyzed with a FACSCalibur flow cytometer (BD). A total of 10,000 events were counted, and labeling data were acquired. Analysis of the data was performed using FlowJo v8 software (BD).

To evaluate the minimal inhibitory activity (MIC) of the salts, a solution containing the same proportion of copper and silver was prepared using pentahydrate Copper sulfate and silver nitrate in sterile water. As a control for these solutions, pure copper and silver salt solutions were prepared, being that the silver solution was proportionally diluted (in comparison with the copper solution) so that its composition was close to that of the alloy. The salt solutions were tested against the pathogen *Escherichia coli* (ATCC 8739). The samples were than serially diluted between concentrations of 450 mmol/L and 3.3 pmol/L. This dilution was caried out using Luria Bertani broth bought from Invitrogen. The final inoculum volume tested was 10^6^ CFU/ mL. The MIC test was then carried out in a 96-well microplate from Corning Incorporated COSTAR^®^ (Corning, NY, USA) at 37 °C for 16 h. The growth was then measured by means of the optical density of the medium using a UV–visible spectrophotometer (Biotek, Winooski, VT, USA) with a fixed wavelength of 600 nm every 10 min.

To measure the MIC of the alloy samples, the samples were dissolved in nitric acid and then diluted with acetate buffer (pH = 7.0) Then, the solution was tested against *Escherichia coli*. The metallic solution was diluted between 8 and 0.06 µg/mL, with a cell concentration of 10^6^ CFU/mL at 37 °C for 16 h.

For metallurgical characterization, the alloy was received as strips and was cut and embedded into resin to obtain images by optical microscopy using an Olympus optical microscope. A D8 Discover diffractometer (by Brucker, Billerica, MA, USA) with Cu radiation was utilized to obtain the X-ray diffraction patterns. An Autolab PGSTAT204 potentiostat/galvanostat from Metrohm (Herisau, Switzerland)was used to carry out the electrochemical characterization using a three-electrode cell. The electrodes used were as follows: a Ag/AgCl reference electrode, a platinum counter-electrode, and the sample as a working electrode. The assays were carried out using saline with 3.5% NaCl and a pH of ~6.3 at room temperature.

The American standard [18] was used as a reference for a quantitative method to assess the treated product’s ability to inactivate H1N1 virus particles at contact times of 1 min, 30 min, and 2 h. Briefly, MDCK cells (Madin–Darby Canine Kidney cells, catalog number CCIAL 068—Instituto Adolfo Lutz) were seeded in 96-well plates. After 24 h, the virus suspension was serially diluted in culture medium and transferred to the 96-well plate. After 1 h of viral adsorption, DMEM high-glucose medium supplemented with 2.5% fetal bovine serum was added to infected wells. The plate was incubated for 72 h. After incubation, the monolayers were fixed and stained with Naphthol Blue Black (Sigma-Aldrich Co., Deisenhofen, Germany) dissolved in sodium acetate–acetic acid. The viral titer is expressed in TCID50/mL and calculated using the Spearman and Kärber algorithm, as described [19].

## 3. Results

Evaluation of the antimicrobial capacity of the copper alloy surface was carried out using a modification made by the group in the ISO 22196/2011 [16], since microbiological test standardization on metal surfaces is still in the early stages. This modification assured that the microorganisms would stay viable and not dry out during the test, as described in the Methods section of this paper. It can be applied to any non-porous material.

The first stress test was carried out using *E. coli* and *S. aureus*; both are model bacteria, as the first is a well-known method of quality checking for water contamination and the latter is a well-known commensal of human skin [20,21,22,23]. Figure 1 shows the results of the stress test. It is possible to notice that both bacteria are no longer viable after 24 h of contact, and *S. aureus* is no longer viable just after 30 m of contact. So, the *E. coli* presents a higher resistance to the copper ion; this is due to the fact that Gram-positive bacteria demonstrate an elevated uptake of metal ions when compared to Gram-negative bacteria due to the presence of glycoproteins, and the presence of both phospholipids and lipopolysaccharides (LPS) diminishes metal absorption [24]. To aggravate this effect, to regulate the periplasmic space, Gram-negative bacteria present transport structures such as porins, efflux pumps, and transenvelope machines, leading to better adaptation to metal-rich environments. This metal transport can lead to higher metal resistance in Gram-negative bacteria [25,26]. This type of behavior is observed in wild strains of metal-resistant bacteria [27,28] and even in normal bacteria regarding metals with micronutrient importance such as Cu [29,30]. These factors allow for the higher viability of *E. coli* during the first couple of hours in contact with the alloy in comparison with *S. aureus*. It is important to note that the positive control is the same for every time stamp because due to the high pathogen concentration of the inoculum, without any biocidal agent, the load extrapolates the colony count per Petri dish.

As mentioned, the most accepted theory regarding the metal surface’s biocidal activity is that the ions released in solution interact with the microorganism’s metabolism, thus killing it [9]. So, to investigate if the addition of silver to the alloy is sufficient to promote higher toxicity, a metallic ion solution from metal sulphate salts was prepared at the same concentration as the alloy. The MIC for *E. coli* in the Cu-Ag solution was determined to be 870 µg/mL. As a reference, the researchers also determined the MIC for a pure Cu solution and a pure Ag solution (with the same concentration as the mixture). For the silver solution, the MIC was determined to be 0.26 µg/mL, and the MIC for the copper solution was 870 µg/mL.

This result shows that silver only increases the antimicrobial activity of the copper in the alloy, not in the solution, as the mixture of Cu^2+^ and Ag^+^ presented the same MIC as the pure Cu^2+^ solution. As the tested alloy is a commercial product, it has trace elements inherent to its metallurgical process. To rule out these trace elements affecting the higher biocidal activity, a sample of the alloy was dissolved in aqueous sulfuric acid and corrected to a neutral pH. The MIC for this solution was determined to be 870 µg/mL, the same as that of a pure copper solution. Therefore, this proves that the alloy’s higher biocidal activity is due to the solid-state interactions between Cu and Ag.

Therefore, it is necessary that we take a metallurgical approach to better understand the effects of the addition of Ag to the system.

Figure 2 shows the morphological characterization of the alloy, and Figure 2a shows the alloy XRD pattern and a typical pattern for pure Cu available from the Inorganic Crystal Structure Database (ICSD) with number 7954. It is important to note that the experimental and the reference diffractograms for copper present the same peaks, indicating that the Ag atoms replace the copper atoms. This is important, as the Cu-Ag alloy presents low solid solubilities [31,32,33]; however, in the studied concentration, the silver appears to substitute copper atoms in the α phase. It is possible to notice a pronounced preferential direction due to the lamination of the alloy during its manufacture, as expected [34,35]. Figure 2b shows the optical microscopy of the etched alloy. Both characterizations show that structurally, the sample is very similar to pure Cu, indicating no significant structural modifications introduced by the silver micro-addition.

To better assess the ion release capability of the alloy, electrochemical assays were conducted for the alloy and the pure metallic elements, thus determining their corrosion resistance. Figure 3a shows the open circuit potential (OCP), which determines the corrosion potential of the material, showing how susceptible the material is to the start of the corrosion process [36]. The result shows that a 0.03% Ag addition promotes a 5% decrease in the OCP. Figure 3b shows the linear polarization measurements obtained, which determine the current density as a function of the applied potential, showing the corrosion rate and resistance tendencies. These can be expressed as the current (I_corr_)and potential of corrosion (E_corr_), respectively, as shown in Table 1. The results shown in Table 1 corroborate previous observations showing an increase in corrosion resistance and indicating a decrease in the corrosion rate.

Figure 4 shows the cyclic voltammetry obtained for the pure metals and the alloy. This measurement is a powerful method of determining the mechanism behind the reduction and corrosion of metals. Figure 4a shows the obtained voltammetry, showing that the profile for the CuAg alloy is very similar to that of the Cu sample. Figure 4b shows the oxidation of the pure Cu and CuAg in more detail. This figure shows that the oxidation peaks are shifted to higher potentials than those verified in the pure Cu. In addition, the current densities of the oxidation peaks in the CuAg alloy decrease compared to the pure copper. Electrochemical measurements indicate that the silver addition at low concentrations induced a higher resistance of copper oxidation in comparison to the pure Cu [32,37,38,39].

To test the effect the micro-addition of Ag, the alloy sample was tested against more resistant pathogens. The contact test was carried out using *P. aeruginosa* and *C. albicans*. The first mentioned pathogen is a well-known opportunistic bacterium that apart from causing serious infections is known for its ability to form biofilms. The second pathogen is also an opportunistic microorganism, being the yeast responsible for causing candidiasis. The contact test results can be seen in Figure 5; these results show that the alloys were effective against *C. albicans* and initially against *P. aeruginosa*. However, *P. aeruginosa* is a famous for its ability to form biofilms; this exopolysaccharide matrix isolates the bacteria from the copper, allowing its growth.

To confirm that the alloys not only make the bacteria on their surfaces unviable but also promote bacterial death, flow cytometry curves were obtained. Figure 6 shows the flow cytometry results obtained for the alloy exposed to *C. albicans* marked with propidium iodide (PI). Figure 6a shows the unmarked viable *C. albicans*, which presents a lower intensity reading of PI; this indicates that the cells are still viable, with a peak of 5 in fluorescence and more than 96% viability. After initial contact, as shown by the curve in Figure 6b, there is a very pronounced decrease in viability and a very high PI count of about 2 × 10^3^; as a result, more than 94% of the cells lost their membrane integrity and were therefore considered dead.

Figure 6c,d show the flow cytometry results obtained after 2 and 24 h of contact with the pathogenic solution. These results show an initial increase in viable cells (at around 40% in Figure 6c) and a further decrease (to 15%) in Figure 6d. At a certain point, the remaining cells are no longer capable of replicating, leading to the low viable count seen in Figure 6d.

Due to the positive results in bacterial inhibition, the researchers also tested the alloy against a virus. The chosen virus was the swine flu virus (H1N1), a subtype of influenza and a model virus due its genetic diversity, human adaptation, pathogenicity, and antiviral response.

To test the alloy, its surface was exposed to a viral solution. This solution was then tested to determine the 50% tissue culture infectious dose (TCID50/mL) by infecting monkey cells to determine the viral load remaining in the suspension.

The tissue culture was carried out with samples of the suspension after 1 min, 30 min, and 2 h of exposure to the alloy surface. These results are shown in Figure 7, which shows that after only one minute of contact, 90% of the viral load was decreased, and after 2 h, 99.999% of the viral load was eliminated.

## 4. Discussion

Both silver and copper are known for their biocidal capabilities and potential as antimicrobial surfaces [40,41,42,43]. Although the addition of silver is a consolidated strategy for promoting the aesthetic and biocidal activity of copper, few studies evaluating the micro-addition of this element to copper can be found in the literature [39]. This study focused on the the micro-addition of Ag to copper to create Ag/Cu alloys.

Besides being a small addition (0.03 wt.%), this modification is enough to enhance bacterial inactivation (Figure 1 and Figure 5). The micro-addition of silver did not modify the material’s morphological properties (Figure 2) but enhanced its corrosion resistance (Figure 3 and Figure 4) due to a slower release of Cu^2+^ ions from the alloy when compared with pure copper. Initially, this may seem like an adverse modification as the ion release occurring in the corrosion process is responsible for the biocidal activity of both copper and silver. However, the slow but constant release of Cu^2+^ is enough to kill the pathogen on its surface.

Flow cytometry results (Figure 6) confirmed these results, showing that the cells remaining after the contact (Figure 6b) try to replicate, initially forming new vi-able cells (Figure 6c). However, the slow and continuous release of copper ions from the alloy keeps killing these new cells, which shows the importance of silver in the alloy’s composition. Therefore, our biocidal and electrochemical results suggest that a slow release of Cu^2+^ due to the presence of silver in the alloy improved the biocidal activity of the material.

These observations show that even in very small concentrations, the addition of silver to copper can enhance its antimicrobial properties, even being able to inactivate model viruses such as the H1N1 virus. This is a very important finding that has been brought back into the limelight by the SARS-CoV-2 pandemic [44,45]. Despite the advances in vaccine and antiviral treatments seen in recent years, there is still an urgent need for the development of new antiviral agents including novel strategies for reducing viral exposure from surfaces. The obtained results demonstrate the capacity of the studied alloy to be used as an effective antimicrobial and antiviral surface.

These results show the importance of a slow and controlled release of copper ions that ensure pathogen death even after long periods. If the exposure is very high and short, the material will kill almost all pathogens, but if the ion release is not constant, the remaining viable cells can replicate. Thus, slow release provoked by the micro-addition of silver ensures a higher availability of ions in the long run (and a greater biocidal effect), thereby keeping the material’s original properties longer, since the process of corrosion consumes the material surface. Regarding the virus, in the first minute of contact, the alloys were capable of reducing the viral load by 90%.

In summary, the addition of silver enhances the material biocidal effect and maintains the product’s mechanical and visual aspects for longer. As this alloy is already fabricated in many forms by metallurgic industries, it is a very compelling material to use as a biocidal surface. In addition, the results show that a small addition of Ag is required to enhance the material’s activity and the alloy’s properties.

## 5. Conclusions

The present study of the Ag/Cu alloy shows that even a small concentration of silver can enhance the biocidal properties of copper while keeping the original material’s mechanical properties. This alloy can slowly release copper ions in a controlled way due to the silver’s ability to protect against copper corrosion, which promotes a constant antimicrobial effect over time. The slow and constant release of Cu^2+^ ions ensures that surviving pathogens are also inactivated and prevents their replication. The results of this study demonstrate the importance of the slow and controlled release of ions in ensuring long-lasting antimicrobial properties. The studied alloy was effective in inactivating the model pathogens *E. coli*, *S. aureus*, and *C. albicans*, representing Gram-negative and -positive bacteria and fungi. The alloy was also capable of inactivating the H1N1 virus in a matter of minutes, showing its potential in pandemic scenarios [44,45]. This functional material is a compelling option as a biocidal surface due to its previously established fabrication in various shapes and its low cost compared to other alloys with higher Ag contents.

## Figures and Tables

**Figure 1 materials-17-00917-f001:**
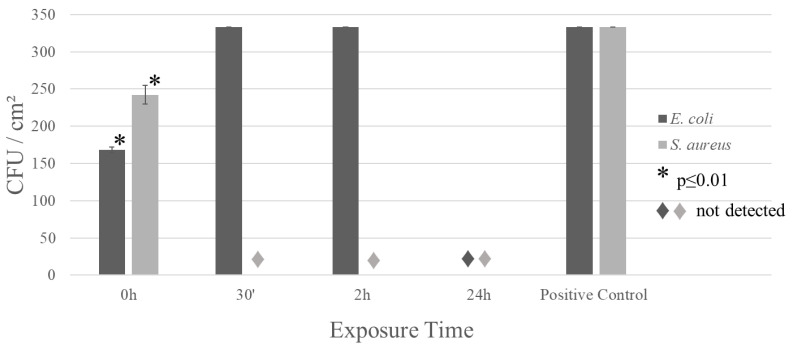
Contact test using *E. coli* and *S. aureus* on the surface of the alloy. All samples were sanded down to 600 grit before the test to expose a fresh surface. * Denotes significant statistical difference according to a one-sample *t*-test between the tested value and positive control, considering *p* ≤ 0.01.

**Figure 2 materials-17-00917-f002:**
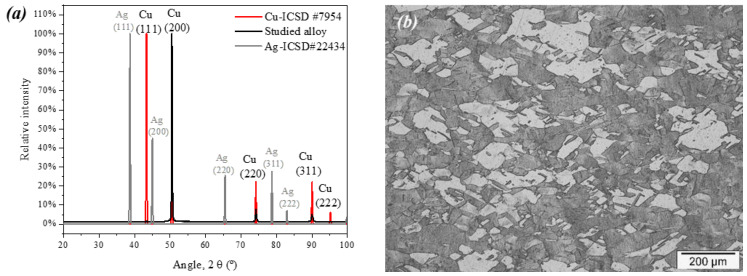
(**a**) XRD patterns of the studied alloy and pure Cu as well as the (**b**) optical microscopy obtained for the alloy.

**Figure 3 materials-17-00917-f003:**
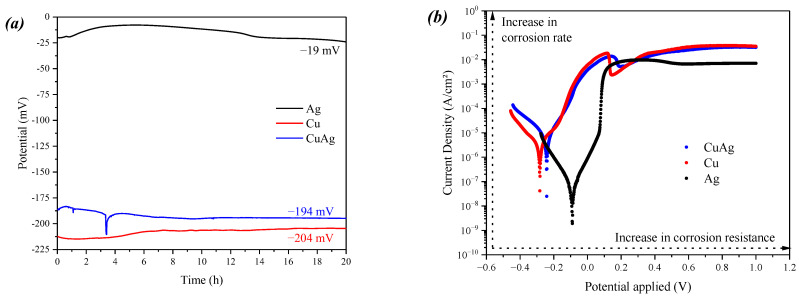
(**a**) Open circuit potential and (**b**) linear polarization for the CuAg alloy and pure Cu and Ag samples.

**Figure 4 materials-17-00917-f004:**
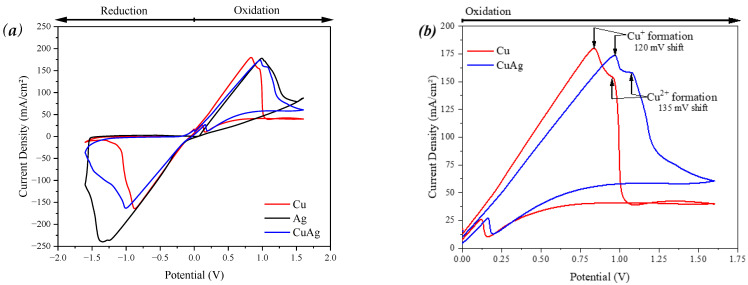
(**a**) Cyclic voltammetry obtained for the studied alloy and the pure metals Cu and Ag, as well as (**b**) the oxidation process for pure Cu and the CuAg alloy studied.

**Figure 5 materials-17-00917-f005:**
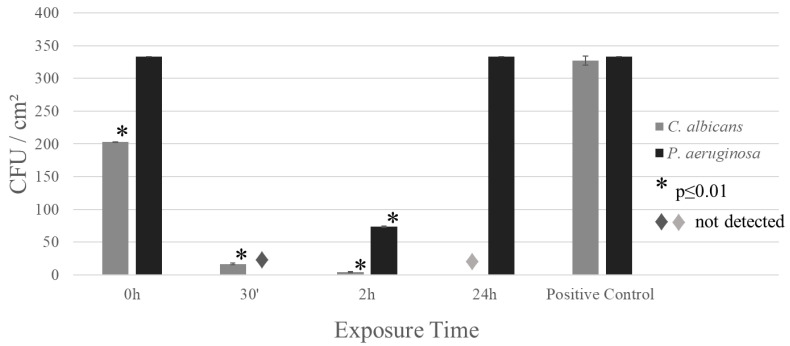
Contact test using *C. albicans* and *P. aeruginosa* on the surface of the alloys. All samples were sanded down to 600 grit before the test to expose a fresh surface. * denotes significant statistical difference in a one-sample *t*-test between the tested value and positive control, considering *p* ≤ 0.01.

**Figure 6 materials-17-00917-f006:**
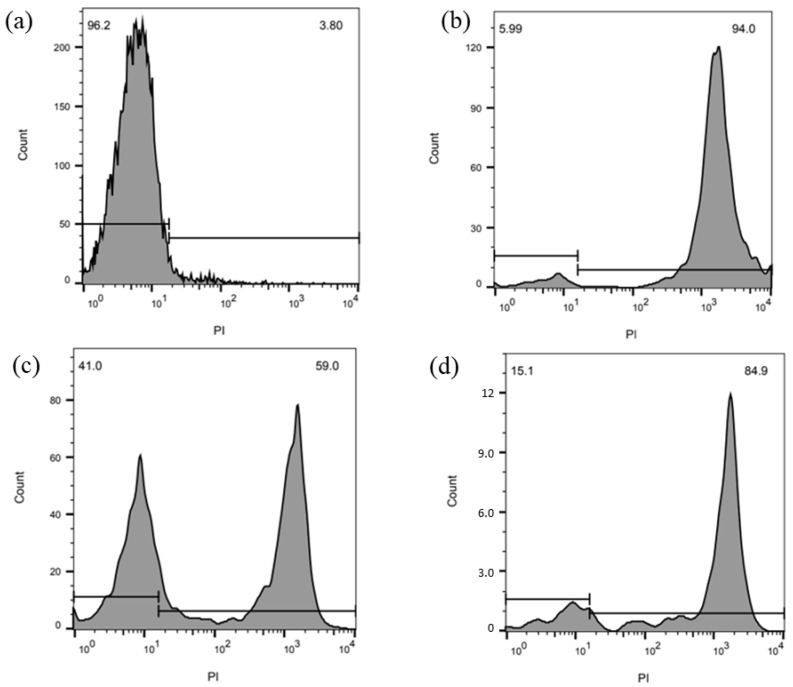
The viability of *C. albicans* was analyzed by labeling with propidium iodide (PI). (**a**) Negative control—not treated and unmarked *C. albicans*; (**b**) *C. albicans* immediately after contact with the alloy, (**b**) 2 h of contact, and (**c**) after 24 h of contact; (**d**) positive control.

**Figure 7 materials-17-00917-f007:**
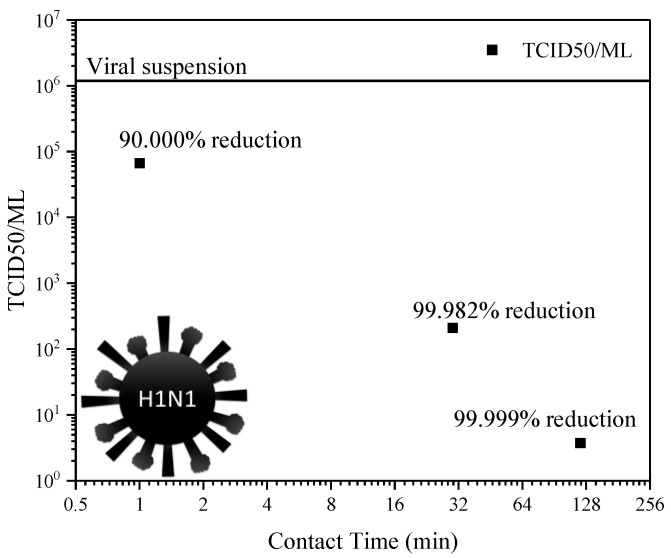
Results from exposure of the alloy sample to the H1N1 virus (HU10680 H1N1 PAN, GenBank Accession No. DQ335993.1).

**Table 1 materials-17-00917-t001:** Corrosion current and potential determined for the pure elements and the studied alloy.

Samples	Ag	Cu	CuAg
I_corr_ (A)	3.46 × 10^−8^	9.57 × 10^−6^	4.42 × 10^−6^
E_corr_ (V)	−0.09	−0.28	−0.24

## Data Availability

Data are contained within the article.

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
