# Peer review of "Micro-Addition of Silver to Copper: One Small Step in Composition, a Change for a Giant Leap in Biocidal Activity"

_materials, 2024, doi:10.3390/ma17040917_

Round 1

Reviewer 1 Report

Comments and Suggestions for Authors

This manuscript provides an investigation into the biocidal properties of Ag/Cu alloy, focusing specifically on the effects of limited Ag additions. The authors employ various methods to examine these behaviors. The manuscript is well-composed and holds promise for publication acceptance. However, there are several areas requiring clarification and additional information from the authors before I can endorse its acceptance.

1.       In the "Materials and Methods" section, the process for creating the Ag/Cu alloy needs to be detailed.

2.       On Line 138, a reference is required to substantiate the claim regarding the effects on bacteria and human skin.

3.       The formatting inconsistency on Line 160 needs rectification, as the initial words are in a different font from the rest of the text.

4.       Figure 2-a should have its peaks indexed. Furthermore, the statement "Ag dissolves into copper" is scientifically inaccurate and needs revision. Please also include the relevant Ag data.

5.       On Line 202, where it is concluded that "low concentrations of silver increase copper's resistance," comparative analysis with other studies is necessary to support or contrast this claim.

6.       In Figure 5, the author should explain the notably large error bar for "C. albicans" under the positive control condition.

Author Response

Dear Reviewer, thank you very much for taking the time to review this manuscript. The complete point-by-point response was uploaded using the journal template, please see the attachment. 

At your disposal, 

Dr. Paganotti 

Reviewer 2 Report

Comments and Suggestions for Authors

The manuscript may interest researchers, but the following key considerations must be addressed in the manuscript for publication:

  1. Figure 1 shows the positive control. Was it measured for both microorganisms at each proposed time? The authors should place the positive control bar at each time.
  2. Should the authors explain why E. coli resist up to 2 hours and S. aureus is inhibited after 30 minutes?
  3. Gram-negatives have a lower amount of peptidoglycan than Gram-positive, explain why in the results section mention that Cu is absorbed mainly in Gram-negative bacteria due peptidoglycan. The authors should explain in more detail the mechanisms of Gram-negative and Gram-positive bacteria, for a better explanation of their results.
  4. The authors do not carry out statistical analysis, it is suggested to add them.
  5. In the discussion, it is suggested to compare results with other authors. The authors do not compare their results with other authors, there are no citations in this section.
  6. Reformulate the conclusions and include the microorganisms and viruses that were tested what the conclusions of this study were and their importance.

Additional commentaries:

Line 96, 115, 121: Eliminate space between the number and °C

Line 97: Eliminate space between 90 and %

Line 120: Italicize scientific name (Escherichia coli)

Line 121: Use only the h for hours as on line 95

Line 155: Place the comma next to and

Line 167: Place the abbreviation XRD on line 167

Line 204: Generate a space between Figure 4 and (a)

Line 246: Use a single decimal as in the rest of the document

Line 248 and 249: Italicize scientific name (C. albicans

Line 308: Place in subscript 2+

Comments on the Quality of English Language

Authors must check the English language in the manuscript

Author Response

(The authors gave the same response as above.)

Round 2

Reviewer 2 Report

Comments and Suggestions for Authors

It must be described that they performed the t test in the methodology.

Line 165-167: Gram-negative bacteria are those that present glycoproteins, phospholipids and lipopolysaccharides in their membrane, and the periplasmic space is also found in Gram-negative bacteria. The information and justification of the results is not good. The authors must describe the mechanisms presented by Gram positive and Gram-negative bacteria that allow E. coli to resist for 2 hours and S. aureus for 30 minutes.

The discussion remains very brief, only 5 articles were added, it is suggested to address the discussion in greater detail in the results obtained in this research.

Line 165: Capitalize the first letter of Gram

Author Response

Dear Reviewer, thank you very much for taking the time to revise once again this manuscript. The complete point-by-point response was uploaded using the journal template, please see the attached file. 

At your disposal, 

Dr. Paganotti 
